# Does digital inclusive finance promote industrial transformation? New evidence from 115 resource-based cities in China

Fei Li[1], Yufei Wu[1], Jinli Liu[2], Shen Zhong[2]*

1 School of Economics, Beijing Technology and Business University, Beijing, China, 2 School of Finance, Harbin University of Commerce, Harbin, Heilongjiang, China

* 102714@hrbcu.edu.cn

**Data Availability Statement:** The data underlying the results presented in the study are available from the Digital finance Research Center of Peking University, as well as"China City Statistical Yearbook"and "China Statistical Yearbook" https://

## Abstract

Industrial transformation (IT) is the inevitable course for the sustainable development of resource-based cities, while digital inclusive finance (DIF) provides essential capital elements for industrial transformation. Based on the panel data of 115 resource-based cities in China from 2011 to 2019, this paper discusses the influence mechanism of digital inclusive finance on industrial transformation from the perspectives of the optimization of industrial structure (OIS) and the rationalization of industrial structure (RIS). The empirical results show that digital inclusive finance has a positive effect on both the optimization of industrial structure and the rationalization of industrial structure. Digital inclusive finance influences industrial transformation through residents' income and technological innovation. In addition, in the analysis of income gap and innovation gap, low-income regions have a better effect on the industrial transformation of industrial structure optimization, while high-income regions have a better effect in manufacturing upgrading, thus resulting in a more significant effect of industrial transformation on the rationalization of industrial structure. Obviously, the development of high-innovation regions has relative advantages with more channels for industrial transformation, which have significant effect of industrial transformation. Therefore, it is necessary to provide differentiated reform on the basis of unified development reform.

## 1 Introduction

Resource-based cities, formed by the exploitation and processing of local natural resources such as mining and forests as the leading industries [1,2], rise with the exploitation of resources and decline with the depletion of resources [3]. According to *Global Mining Industry 2019*, mineral resources account for more than 80% of resources consumed by human beings, and the processing and exploitation of mineral resources are the leading industries in resource-based cities. Therefore, these resource-based cities play an important role in the overall economic development [4]. However, under the dual pressure of economic development and social demand, resource-based cities have shown an excessive dependence on resources and a boom in primary processing manufacturing [5]. It should note that the long-term

idf.pku.edu.cn/index.htm https://data.cnki.net/
Yearbook/Single/N2021110004 https://data.cnki.
net/yearbook/Single/N2021050059.

**Funding:** This study was funded by the
Heilongjiang Philosophy and Social Sciences
Project (21JYE396).

**Competing interests:** The authors have declared
that no competing interests exist.

extensive development model has brought a series of problems, for example, the exhaustion of urban leading industries has led to negative economic growth [6], declining of employment rate [7], diminishing marginal returns [8], imbalance of industrial structure [9] and other problems. In particular, the problem of resource curse is prominent in resource-based cities [10]. Therefore, it is necessary to solve the above problems through the realization of industrial transformation, because industrial transformation can transfer low productivity sectors to high productivity sectors, thus forming technology-intensive and knowledge-intensive industries [11]. There is no doubt that changing the industrial structure and guiding the industrial transformation are the driving force for the sustainable development of resource-based cities [12].

Capital is the necessary core factor in the process of industrial transformation, and finance is the core of capital allocation, which realizes the effective allocation of inter-industry resources [13]. Financial development provides appropriate and effective financial services at affordable costs by strengthening financial system construction and improving financial infrastructure. However, financial development needs to be achieved through the establishment of physical places and equipment, which exists a phenomenon of insufficient effective supply [14]. The integration of digital technology and financial services has overcome this shortcoming, it is obvious that digital inclusive finance eliminates geographical restrictions and has the advantages of low threshold and low cost, making it possible for people in some areas to access financial services through the Internet [15,16]. In addition, digital inclusive finance makes financial services more convenient, and customer coverage becomes more extensive [17]. In summary, it is of theoretical and practical significance to study the role of digital inclusive finance of resource-based cities in industrial transformation, this paper will make an in-depth study of this problem.

The second part of this paper makes a theoretical analysis on the mechanism of digital inclusive finance on industrial transformation. The third part puts forward the research hypothesis. The fourth part introduces the selection of variables and models. The fifth part carries on the empirical analysis, it further analyzes the transmission mechanism of digital inclusive finance and industrial transformation in resource-based cities, and divides them into groups according to the income level of residents and the degree of innovation, at the same time, it also solves the endogenous problems with instrumental variables. The sixth part conducts a discussion. The seventh part provides conclusions and policy recommendations.

## 2 Literature review

The development of resource-based cities has attracted much attention from scholars. Auty [18] put forward the concept of " resource curse", which means that countries and regions with relatively rich resources develop slowly, and resources cannot be fully utilized. Instead, they have a high dependence on natural resources, resource endowments have become a stumbling block to economic development. Many studies have conducted in-depth studies based on this issue, the emphasis is on the difficulties such as the decline of natural resources (Zeng et al., 2016), the crowding-out effect on other production factors [19], the imbalance of economic structure [20], and the increase of unemployment rate [21]. In recent years, people have tried to find the way of "resource gospel" and paid more attention to the transformation of resource-based cities. The development of underdeveloped industries in resource-based cities has lost the attractiveness of foreign investment, thus leading to the deterioration in terms of trade and the phenomenon of "Dutch disease" to further promote the vicious circle of industrial development in resource-based cities [22]. The industrial transformation of resource-based cities has become an urgent problem to be solved [23,24].

At present, the research on industrial transformation is mainly divided into two directions. One is about the path of industrial transformation and research methods, the other is the research on the influencing factors of industrial transformation. On the research of industrial transformation path, Humphrey and Schmitz [25] proposed to incorporate industrial transformation and upgrading in the framework of global value chain analysis, starting from the global value chain to study the mechanism of industrial upgrading. Pólvora et al [26] proposed to use mixed desktop research with qualitative methods under the blockchain system to study the challenges and opportunities faced by relevant industrial transformation. Lü et al. [27] and Busch et al. [28] explored green industrial transformation schemes, combined with socio-economic and policy objectives in the context of global environmental change. At present, scholars' measurement indicators of industrial transformation performance are mainly divided into two aspects. On the one hand, it simplifies the industrial transformation and upgrading, by constructing an industrial transformation performance evaluation system to make a measurement. Zhou et al. [29] took the tertiary industry with higher added value as the evaluation index of special type of industrial, and measured its performance by the change rate of its proportion in GDP. On the other hand, it carries on the comprehensive measurement to the industry types. Song et al. [30] used grade coefficient of industrial structure as a reference to measure industrial transformation, which reflects the evolution of industrial structure. Zhou et al. [31] took into account the heterogeneity of the three industries, and used the Theil index to give different weights to different industries to consider industrial optimization. Previous studies are mostly limited to single-index studies, while ignoring the analysis of positive and negative indicators of industrial transformation.

On the discussion of the influencing factors of industrial transformation, many factors have been studied, such as capital market, environmental regulation, government policies and other factors. Wang et al. [32] showed that there is a certain correlation between capital market and industrial transformation. On the basis of considering the factors of economic development, Lai et al. [33] studied the "inverted u" effect of market segmentation on industrial transformation, and mainly analyzed the relationship between market segmentation and sustainable provincial market segmentation from the perspective of industrial transformation of environmental protection. Zhang et al. [34] employed the Propensity Score Matching-Difference in Difference to investigate the impact of sustainable development policies on carbon emissions in resource-based cities. Fan and Zhang [4] pointed out that the key point of industrial transformation is how to get rid of the resource-based industrial chain and achieve sustainable transformation. Through the comparison between the implementation of Sustainable Development Planning of National Resource—Based Cities and the implementation of government planning, it is found that the implementation of government planning has a positive impact on industrial production. Brandt and Thun [35]; Lin and Zhou [36] highlighted the impact of government policies on industrial transformation. Case studies showed that supply and demand constraints have a restrictive impact on the expertise and availability of resources required for upgrading products. In addition, in recent years, scholars found that education investment [37] and human capital [38] are important factors affecting industrial transformation. In the study of influencing factors of industrial research, more attention is paid to the impact of policy system and social culture, while ignoring the role of financial markets.

Financial development is the main influencing factor of industrial transformation [39]. Arjun et al. [40] empirically studied the long-term and short-term dynamics of financial development on manufacturing development in the secondary industry through ARDL marginal test and VECM causality test, and provided evidence for the leading direction of financial supply and demand. Ferraz and Coutinho [41], from the bank point of view, took the Open Bank of Brazil as an example and found that the bank's financial policies drive the renewal of

facilities, the proportion of employment and the coverage of services, which are stable drivers of industrial economic transformation. Jiang et al [42] explored the nonlinearity of industrial transformation from the perspective of absorptive capacity with financial development as the double threshold. Xu and Tan [43] established a spatial econometric model, by considering financial development in terms of scale, structure and efficiency, to effectively promote the process of industrial transformation in terms of scale efficiency and reasonable structural arrangement. Wang and Wang [44] determined the correlation between green finance and industrial upgrading based on the system GMM model, and clarified the differences in the impact on the three industries. Previous studies have paid more attention to the impact of financial development on industrial structure, while ignoring the background of big data economy. In theory, digital inclusive finance is an extension of traditional finance and an important source and growth point of traditional finance. Specifically, in terms of the coverage areas, the traditional financial services need to increase coverage by setting up institutional outlets, but the high cost of institutional outlets makes it difficult for traditional financial services to penetrate into the economically backward areas. It should note that the cross-border integration of digital technology and financial services overcomes this drawback, and customers can obtain the financial services they need through terminal equipment, such as computers and mobile phones [45]. From the perspective of the covered social groups, the product innovation of digital inclusive finance lowers the user threshold, showing the trend of popularization for financial services and the non-exclusive nature of digital inclusive finance. These can satisfy the needs of small, medium and micro enterprises and low-income groups, thus reflecting the proper meaning of digital inclusive finance.

In summary, although scholars have carried out extensive research on the impact of financial development on industrial transformation, the following aspects still need to be improved: (1) The existing research is limited to the impact of traditional finance on industrial transformation, ignoring the role of digital inclusive finance guaranteed by digital technology in the new era. (2) Most of the literature ignored the internal analysis of industrial transformation, and did not further analyze the internal mechanism of industrial transformation. (3) Scholars pay more attention to the coordination, causality and linear relationship between the two, but ignore the transmission mechanism between them, that is, what kind of intermediate channels financial development plays a role in industrial transformation.

This study may have the following contributions. (1) This article uses the data jointly prepared by the Digital Finance Research Center of Peking University and Ant Technology Group to analyze the impact mechanism of digital inclusive finance on industrial transformation on the digital finance platform for the first time. (2) This article analyzes the positive and negative indicators of digital inclusive finance on industrial transformation from both theoretical and empirical directions for the first time, and deeply analyzes the important influence of digital inclusive finance on the industrial transformation of resource-based cities. The mechanism is shown in Fig 1. (3) This article uses residents' income and technological innovation as intermediary transmission mechanism, and conducts further grouping analysis according to residents' high income, low income, high innovation and low innovation.

## 3 Research hypotheses

Hypothesis 1: Digital inclusive finance promotes industrial transformation.

With the support of emerging technologies, digital products are gradually changing the market layout and reshaping industrial structure. There is no doubt that the popularization of digital finance provides conditions for creating market value and reshaping business models [46]. The combination of big data and finance has changed the layout of consumers, producers

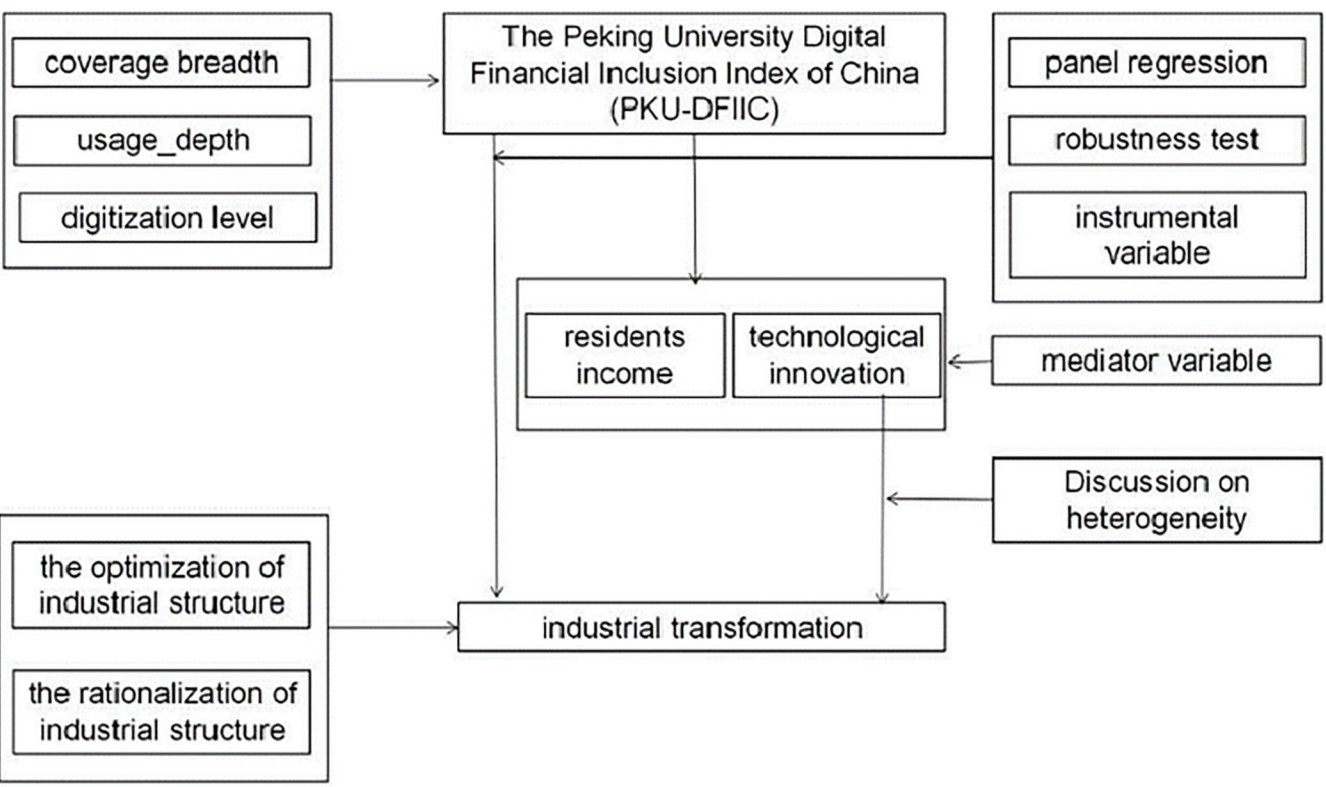

**Fig 1. Mechanism of the effect of digital inclusive finance on industrial transformation.**

and the market. Consumer demand-oriented market model has promoted the pace of social progress and inevitably accelerated the trend of industrial transformation [47]. By enhancing the temporal and spatial permeability, convenience and inclusiveness, digital inclusive finance has achieved a high degree of coordination between big data and financial functions. In other words, digital inclusive finance can increase information transparency, reduce information asymmetry between supply and demand, so as to reduce search matching costs and improve the efficiency of financial services. Digital inclusive finance expands the financial service boundary on the basis of reducing transaction costs and expanding coverage rate to improve financial availability. This intensifies the interest rate adjustment function of the credit market, optimizes the credit investment structure, and realizes industrial transformation and upgrading [48]. Therefore, it has improved the ability of financial services to influence the economy on a larger scale, the market development trend and the help of digital finance can better achieve the goal of industrial transformation.

Hypothesis 2: Digital inclusive finance affects industrial transformation and upgrading by increasing residents' income, that is, there is a positive mechanism path of "digital Inclusive finance—residents income—industrial transformation".

Chinese economy has entered into a new normal, but the liquidity constraints of income, imperfect security system and income inequality are the main problems faced by the government. Financial development can reasonably allocate resources to achieve intertemporal smoothing of residents' income, so as to hedge liquidity constraints of residents and drive stable consumption demand [49]. With the further integration of Internet technology and financial development, digital inclusive finance affects residents' income in many aspects. Firstly, digital inclusive finance matches the supply side and demand side of online credit with each

other [50]. With the various P2P credit platforms, represented by Alipay and Yulibao, lowers the barriers to entry, lending and financing channels have been expended to alleviate the liquidity of residents ' income. Secondly, it also broadens the investment channels for residents to make small investment, increases the return on investment of families. The combination of inclusive finance and digital technology can reduce financial exclusion, thereby increasing the income of residents, especially the income of low-income groups [51]. At the same time, the fast-developing digital financial platform has greatly reduced the "leather cost" [52] and "menu cost" [53] of financial services, and improved the efficiency of financial services for residents. In addition, the emergence of digital finance not only promotes the development of traditional insurance industry, but also drives the development of Internet insurance companies, which breaks the geographical barriers of traditional insurance, so as to increase the stability and availability of insurance, reduce the uncertainty losses. Obviously, the increase in residents' income has stimulated residents' consumption. According to 'consumer theory', the market realizes the transformation of industrial structure according to consumer preferences and expectations. Therefore, with the increase of consumer income and residents' consumption expectations, there is a higher demand for sustainable production of products, leading to industrial upgrading.

Hypothesis 3: Digital finance affects the optimization and rationalization of industrial structure by improving technological innovation. That is, there is a positive mechanism path of "digital financial development—technological innovation—industrial transformation".

The friendliness of the financial system to technological innovation affects its innovation activities and the nature of innovation [54]. However, the lack of traditional financial supply has greatly inhibited the development of innovation activities, so digital inclusive finance provides supplementary support for technological innovation [55]. Digital inclusive finance has realized the optimal allocation of financial resources, improved the passive position of enterprises, improved the relationship between banks and enterprises, and promoted the two sides to make two-way choices. This "partnership" relationship enables enterprises to obtain market initiative. In addition, under the condition of digital economy, digital inclusive finance breaks through the time lag and spatial difference between traditional finance and enterprises, and realizes the interconnection and sharing of various information based on the big data platform. Its development will have a direct spillover effect on other industries. In addition, digital inclusive finance solves the asymmetry and opacity of information, improves the matching efficiency of capital supply. It solves the long-term and short-term capital turnover problems of innovation companies, it is obvious that more abundant cash flow will effectively alleviate the liquidity of corporate funds, and provide necessary financial support for technological progress [56]. At the same time, the richness of overall resources in the industry has also brought about intensified competition in the industry. This digital inclusive competition effect will force enterprises to carry out technological innovation to obtain new competitive advantages [51]. With the savings mobilization function of digital inclusive finance, the ability of technological innovation has been improved. Technology is the main driving force of development. The reform of technological innovation drives the transformation of the industrial structure, thereby realizing industrial optimization and promoting industrial transformation.

## 4 Research area, model and method

### 4.1 Research area

Resource-based cities are important guarantee bases for resource strategies, and their development is the potential and stable driving force of China 's economic growth. In 2013, the State Council announced *National sustainable development planning of resource-based cities* which

identified 262 resource-based cities, including 126 prefecture-level administrative regions, 62 county-level cities, 58 counties (autonomous counties and forest areas) and 16 municipal districts. However, due to the insufficient scale and single industrial structure of county-level cities, counties and administrative regions, it is easy to produce resource squeeze effect, therefore they are not used as a reference. Considering the availability and integrity of data, this paper takes 115 resource-based cities with complete data covering 24 provinces in China as samples for conducting research.

## 4.2 Basic measurement model

Based on the panel data of 115 resource-based cities in China from 2011 to 2019, this paper empirically analyzes the impact of digital inclusive finance on industrial transformation, and establishes the following basic measurement model:

$$OIS_{it} = \alpha_1 + \beta_1 DIF_{it} + \delta_1 X_{it} + \varepsilon_{it} \tag{1}$$

$$RIS_{it} = \alpha_2 + \beta_2 DIF_{it} + \delta_2 X_{it} + \varepsilon_{it} \tag{2}$$

Where, i and t represent the region and time respectively, OIS and RIS are explained variables of the optimization of industrial structure and the rationalization of industrial structure, DIF is digital inclusive finance, $\beta_1$ represents the impact of DIF on the optimization of industrial structure, $\beta_2$ represents the impact of DIF on RIS, X is the control variable, $\delta$ represents the coefficient set of control variables, $\varepsilon$ is the disturbance term. However, the basic measurement model assumes that the regression equation of each city is uniform, that is, there is no heterogeneity between each city. In fact, there are differences in natural endowments and social attributes of each city, and the individual effect model may be random effect or fixed effect. The random effect model is as follows:

$$OIS_{it} = \beta_3 DIF_{it} + \theta_1 X_{it} + \mu_i + \varepsilon_{it} \tag{3}$$

$$RIS_{it} = \beta_4 DIF_{it} + \theta_2 X_{it} + \mu_i + \varepsilon_{it} \tag{4}$$

The explanatory meaning of each variable is the same as that of the explanatory variable in the benchmark regression model. Supposing that $\mu_i$ the individual effect which does not change with time. The fixed effects model supposes that $u_i$ is related to one or more explanatory variables. The random effects model assumes that $\mu_i$ is independent of the explanatory variables, that is, the observed characteristics of individuals are independent of the explanatory variables. In order to test whether $\mu_i$ is related to explanatory variables, the Hausman test is introduced to select the optimal mode.

## 4.3 Variables and data description

**4.3.1 Explanatory variables.**   With regard to the measurement of digital inclusive finance (DIF), the existing literature mainly measures the overall level of digital inclusive finance by constructing indexes, which including three ideas. The first is based on questionnaires and on-site survey indexes, such as the World Bank's Global Findex index [57]. The second is to construct indexes through text analysis. Zhang and Liu [51] shows that the development of inclusive finance depends on the number and coverage of financial institutions, and it is measured by the availability of financial services, the availability of financial products, the normative method and digitalization of financial services products. The third is the construction of indicators through underlying transactions, such as the Digital Inclusive Finance Index and Global

Partnership for Financial inclusion (GPFI), which provide many specific variables for DIF. However, some variables change every three years and the data lacks completeness and continuity [58]. Digital Inclusive Finance Index is an index developed by the Digital Research Center of Peking University based on Ant Financial to reflect digital development [59]. The index can be subdivided into coverage breadth, usage depth, and digitization level of inclusive finance. In addition, usage depth index also includes payment, credit, insurance, investment, monetary fund and other business classification indexes, including provincial, municipal, and county levels. Comparing the advantages and disadvantages of the existing indexes, the explanatory variable DIF in this paper is measured by the digital inclusive financial index and studied by its subdivision index. The index system is shown in Table 1.

**4.3.2 Explained variables.** Industrial transformation (IT). Industrial transformation is the effect achieved by the coordination between industries and reasonable proportions of various sectors of the industry, that is, the process of coordination and unification of the optimization of industrial structure and the rationalization of industrial structure [36]. From a dynamic perspective, this paper takes the optimization of industrial structure and the rationalization of industrial structure as explanatory variables.

Optimization of industrial structure (OIS). The optimization of industrial structure is mainly the process of industrial structure evolution, and its significant feature is the obvious increase in the proportion of tertiary industry. Therefore, this paper selects the proportion of GDP of tertiary industry to measure the transformation of industrial structure. If OIS is in a rising state, it will prove the industrial structure is in a positive transformation.

Rationalization of industrial structure (RIS). The rationalization of industrial structure refers to the aggregation quality between industries, which reflects the degree of resource utilization and the coordinated distribution between industries. This paper uses the proportion of mining industry and manufacturing industry to measure the rationalization of industrial structure.

**4.3.3 Control variable.** This paper employs several variables that will affect IT as control variables.

Government regulation (GR). Government regulation policies have a great impact on the transformation of industrial structure [60]. The improvement of industrial structure optimization needs government support, which can break the bottleneck of industrial development. The main form of government regulation is the government's fiscal expenditure, this paper uses the ratio of local fiscal expenditure to regional GDP to represents the degree of government regulation.

Regional openness (RO). Regional openness is measured by trade openness [61], it is expressed in terms of the actual use of foreign capital that year in regional GDP. Due to the differences in industrial form and industrial development level of each country, according to factor endowment theory, IT direction and use efficiency exist some differences when implementing foreign trade opening.

Economic development level (ED). The study uses per capita GDP (GDPP) to represent the economic and wealth situation [62]. The industrial transformation of a country or region will change with the continuous advancement of economic development. When the per capita GDP is higher, the city will have more resources and income, which plays a positive role in promoting IT. In order to be more persuasive and comparable, this article uses 2011 as the base period, expressed at constant prices.

Capital investment (CI) is measured by the ratio of capital stock to GDP of the year [63]. The capital stock adopts perpetual inventory method, it is calculated by using the annual fixed asset investment and the formula $K_{i,t} = (1-\delta)K_{i,t}-1+It/\varphi_{it}$, where $K_{i,t}$ is the capital stock. $\delta$ is an

**Table 1. Digital inclusive financial indicator system.**

| First-level dimension | Second-level dimension | | Specific indictors |
|---|---|---|---|
| Coverage breadth | Account coverage rate | | Number of Alipay accounts per 10,000 people |
| | | | Alipay card user ratio |
| | | | Average number of bank cards per Alipay account bound |
| Usage depth | Payment transactions | | Average number of bank cards per Alipay account bound |
| | | | Per capita payment amount |
| | | | Number of active users with high frequency (50 times per year and above) Accounted for one times per year and above |
| | Money fund business | | Number of balances purchased per capita |
| | | | Amount of balance treasure purchased per capita |
| | | | Number of people who buy Yu 'e Bao per 10,000 Alipay users |
| | Credit operations | Personal consumption loan | Number of Internet consumer loans per 10,000 adult Alipay users |
| | | | number of loans per capita |
| | | | Per capita loan amount |
| | | Small and micro operators | Number of Internet micro business loan users per ten thousand Alipay adult users |
| | | | Number of loans per household of small and micro operators |
| | | | Average loan amount for small and micro operators |
| | Insurance business | | Number of insured users per 10,000 Alipay users |
| | | | Per capita insurance pens |
| | | | Per capita insurance amount |
| | Investment business | | Number of Internet investors per million Alipay users |
| | | | Number of investments per capita |
| | | | Per capita investment amount |
| | Credit business | | Natural person credit per capita call times |
| | | | Number of credit-based service users per 10,000 Alipay users (including finance, accommodation, travel, social networking, etc.) |
| Digitization level | Mobility | | The proportion of mobile payment pens |
| | | | Proportion of mobile payments |
| | Facilitation | | Average loan interest rate of small and micro operators |
| | | | Personal average loan interest rate |
| | Credit-oriented | | Proportion of Huabei payment pens |
| | | | Payment ratio |
| | | | Percentage of Sesame Credit Deposit Free Pens (total deposit requirement) |
| | | | Proportion of Sesame Credit Free Deposits (total deposit requirement) |
| | Facilitation | | Proportion of pens paid by user 2D code |
| | | | Proportion of amount paid by user 2D code |

annual depreciation rate, which is 9.6%, It is fixed asset investment, $\varphi_{it}$ is the cumulative capital price index for provinces.

Population density (PD) is the number of population per unit area. Population provides labor productivity for social development [64], it is the main force of IT, population aggregation drives the transformation of industrial structure.

## 4.4 Data source

As mentioned above, in this paper, the data comes from the Digital finance Research Center of Peking University, as well as *China City Statistical Yearbook* and *China Statistical Yearbook*. Statistical description of each variable is shown in Table 2.

**Table 2. Descriptive statistics of variables.**

|  | Obs | Mean | Std | Min | Max |
|---|---|---|---|---|---|
| OIS | 1035 | 3.629 | 0.247 | 2.317 | 4.321 |
| RIS | 1035 | 3.451 | 1.804 | -2.753 | 8.124 |
| DIF | 1035 | 4.954 | 0.520 | 3.060 | 5.650 |
| DR | 1035 | 7.490 | 0.637 | 5.857 | 13.720 |
| RO | 1035 | 4.185 | 1.639 | -4.613 | 7.895 |
| ED | 1035 | 10.247 | 0.545 | 8.734 | 12.261 |
| CI | 1035 | 2.326 | 0.053 | 2.167 | 2.506 |
| PD | 1035 | 5.401 | 0.941 | 2.270 | 6.946 |

## 5 Empirical analysis

### 5.1 Base panel regression

Since IT contains two dimensions, the next step is to take OIS and RIS as the explained variables to analyze the influence of each factors on them. In Table 3, model (1), model (2) and model (3) are respectively expressed as mixed OLS, fixed effect and random effect of OIS. Model (4), model (5) and model (6) are respectively expressed as mixed OLS, fixed effect and random effect of RIS. According to the results of F test, LM test and hausman test (Hausman, 1978), it shows that the fixed effect is the optimal model in OIS, and the random effect is the optimal model in RIS. As shown in Model (1), OIS is a positive indicator of industrial upgrading. The coefficient of DIF is significantly positive at 1% significance level. When the coefficient of OIS increases by 1%, DIF will increase by 0.251%, indicating that the higher the level of DIF, the better the financial service and the more beneficial to the tertiary industry. From

**Table 3. Basic regression results.**

|  | (1) | (2) | (3) | (4) | (5) | (6) |
|---|---|---|---|---|---|---|
| DIF | 0.251*** (21.22) | 0.224*** (26.08) | 0.240*** (28.79) | -0.564*** (-5.29) | -0.468*** (-12.01) | -0.456*** (-11.80) |
| DR | 0.039*** (3.63) | -0.003 (-0.29) | 0.013 (1.41) | 0.109 (1.13) | 0.037 (0.83) | 0.033 (0.76) |
| RO | 0.017*** (4.18) | 0.004 (0.94) | 0.006 (1.44) | -0.094** (-2.56) | -0.032 (-1.64) | -0.033* (-1.70) |
| ED | 0.053 (0.18) | -0.835* (-1.67) | -0.549 (-1.30) | 4.166 (1.60) | -2.691 (-1.18) | -1.960 (-0.89) |
| CI | -1.769 (-0.59) | 4.646 (0.90) | 3.395 (0.78) | -41.335 (-1.53) | 25.711 (1.09) | 18.723 (0.82) |
| PD | -0.013* (-1.78) | -0.128* (-1.66) | -0.017 (-1.09) | -0.195*** (-3.02) | 0.642* (1.84) | -0.120 (-082) |
| N | 1035 | 1035 | 1035 | 1035 | 1035 | 1035 |
| $R^2$ | 0.377 | 0.590 | 0.578 | 0.058 | 0.144 | 0.139 |
| F test |  | 12.85*** |  |  | 74.06*** |  |
| LM test |  |  | 1147.16*** |  |  | 3238.06*** |
| Hausman test |  | 52.02*** |  |  |  | 8.31 |

Notes

* p<0.1

** p<0.05

*** p<0.01 the bracket represents the t value.

**Table 4. Regression results of decomposition items.**

|  | (7) | (8) | (9) | (10) | (11) | (12) |
|---|---|---|---|---|---|---|
| Coverage_breadth | 0.200***<br>(24.87) |  |  | -0.416***<br>(-11.71) |  |  |
| Usage_depth |  | 0.225***<br>(26.74) |  |  | -0.402***<br>(-10.21) |  |
| Digitization_level |  |  | 0.153***<br>(19.13) |  |  | -0.364***<br>(-11.24) |
| DR | -0.006<br>(-0.63) | 0.004<br>(0.40) | 0.004<br>(0.38) | 0.040<br>(0.90) | 0.018<br>(0.40) | 0.022<br>(0.50) |
| RO | 0.004<br>(0.94) | 0.003<br>(0.73) | 0.002<br>(0.41) | -0.032*<br>(-1.71) | -0.028<br>(-1.42) | -0.030<br>(-1.55) |
| ED | -0.870*<br>(-1.70) | -1.022**<br>(-2.07) | -0.864<br>(-1.54) | -2.049<br>(-0.93) | -1.854<br>(-0.80) | -2.450<br>(-1.11) |
| CI | 4.302<br>(0.81) | 7.179<br>(1.40) | 4.354<br>(0.75) | 20.776<br>(0.91) | 16.850<br>(0.70) | 23.971<br>(1.05) |
| PD | 0.129*<br>(1.65) | -0.093<br>(-1.22) | -0.091<br>(-1.06) | -0.103<br>(-0.67) | 0.533<br>(1.50) | -0.118<br>(-0.77) |
| N | 1035 | 1035 | 1035 | 1035 | 1035 | 1035 |
| $R^2$ | 0.574 | 0.599 | 0.490 | 0.136 | 0.110 | 0.128 |
| F test | 12.46*** | 13.43*** | 10.79*** |  | 69.64*** |  |
| LM test | 1052.86*** | 1250.66*** | 872.11*** | 3252.25*** | 3133.94*** | 3251.91*** |
| Hausman test | 75.93*** | 37.94*** | 83.22*** | 8.55 | 11.81* | 7.98 |

Notes

* $p<0.1$

** $p<0.05$

*** $p<0.01$ the bracket represents the t value.

the perspective of control variables, due to the "leading role" and "demonstration effect", the level of economic development and population density have a significantly positive impact on IT, which plays a significant role in promoting the industrial upgrading of cities. In addition, RIS is a negative indicator of IT, and DIF is significantly negative at 1% significance level. For every 1% increase in the coefficient of DIF, RIS will reduce by 0.456%, which proves that the promotion of DIF and the upgrading of the mining industry promote the overall IT, and verifies Hypothesis 1.

In order to further analyze the impact of DIF on industrial upgrading, this paper refers to the method of Guo Feng (2020), and further considers the subdivision indicators of digital inclusive finance, namely coverage breath, usage depth and digitization level. In Table 4, columns (7), (8) and (9) respectively represent the regression results of OIS with the segmentation index as the core explanatory variable. The results show that the coverage breath, usage depth and digitization level have a positive effect on industrial upgrading. For every 1% increase in the subdivision index coefficient, the corresponding industrial transformation coefficients will increase by 0.2%, 0.224% and 0.153% respectively. Columns (10), (11) and (12) respectively represent the regression results of RIS with the subdivision index as the core explanatory variable. The results show that DIF has an inhibitory effect on RIS, indicating that it plays a negative role in stimulating IT. For every 1% increase in the subdivided index coefficient, the corresponding industrial transformation coefficients will decrease by 0.416%, 0.402% and 0.367%, respectively. With the increase of the coverage of DIF, the supply of digital financial services ensures that users receive corresponding services to a greater extent. With the enrichment of third-party payment functions, third-party payment has become an important

financial management and financing channel, which accelerates the flow of funds to promote the development of industry. With the continuous expansion of the deep use of DIF, payment services, monetary fund services, credit services, insurance services, and investment services have been continuously improved and the degree of facilitation has been enhanced, which improves the capital investment and capital mobility of the R & D team and ensures the innovation efficiency and thus affects IT. The digitization of DIF realizes the informatization, benefit and mobility of finance, which makes financial services become more convenient, lower cost, higher credit degree. Moreover, capital value has been fully reflected and industrial structure has been improved. At the same time, it can be seen that the absolute value of the rationalization coefficient of the industrial structure is significantly higher than the coefficient of the advanced industrial structure, which indicates that DIF and its subdivision indicators have more obvious effects on the transformation of the mining industry.

### 5.2 Robustness test

In order to test the stability of the empirical results, this paper adopts the method of exchanging data to conduct robustness test. Table 5 uses the proportion of employees in the tertiary industry to represent OIS. Columns (13), (14) and (15) are the mixed OLS, fixed effect and random effect when DIF is used as the core explanation. Columns (16), (17) and (18) are listed as regression results of the sub-items of DIF. The results show that the coefficient only has a little change, neither direction nor significance have changed, which verifies the effectiveness of the empirical results.

**Table 5. Stability test results.**

|  | (13) | (14) | (15) | (16) | (17) | (18) |
|---|---|---|---|---|---|---|
| DIF | 0.049*** (3.79) | 0.048*** (6.70) | 0.047*** (6.80) |  |  |  |
| Coverage_breadth |  |  |  | 0.041*** (6.16) |  |  |
| Usage_depth |  |  |  |  | 0.055*** (7.73) |  |
| Digitization_level |  |  |  |  |  | 0.030*** (4.97) |
| DR | -0.016 (-1.37) | 0.020** (2.45) | 0.015* (1.87) | 0.020** (2.38) | 0.021*** (2.60) | 0.022*** (2.64) |
| RO | -0.016*** (-3.61) | -0.013*** (-3.64) | -0.014*** (-3.90) | -0.013*** (-3.64) | -0.013*** (-3.68) | -0.014*** (-3.75) |
| ED | 0.484 (1.54) | -1.497*** (-3.57) | -1.065*** (-2.79) | -1.522*** -3.62) | -1.496*** (-3.60) | -1.532*** (-3.60) |
| CI | -7.500** (2.31) | 13.717*** (3.16) | 8.971** 2.28) | 13.800*** (3.17) | 13.935*** (3.24) | 13.912*** (3.16) |
| PD | -0.044*** (-5.71) | 0.040 (0.62) | -0.044** (-2.43) | 0.041 (0.63) | 0.0439 (0.69) | 0.050 (0.76) |
| N | 1035 | 1035 | 1035 | 1035 | 1035 | 1035 |
| $R^2$ | 0.324 | 0.212 | 0.209 | 0.206 | 0.224 | 0.195 |
| F test |  | 26.83*** |  | 26.65*** | 27.26*** | 26.22*** |
| LM test |  |  | 2198.57*** | 2190.95*** | 2210.86*** | 2174.18*** |
| Hausman test |  | 13.13** |  | 12.51*** | 15.26** | 12.01* |

Notes

* $p<0.1$

** $p<0.05$

*** $p<0.01$ the bracket represents the t value.

## 5.3 Endogenous problems

Although the benchmark panel regression model has proved the positive effect of digital inclusive finance on industrial transformation, industrial transformation may also have an impact on digital inclusive finance. For regions with complete industrial structure and high level of economic development, the integrity of their industrial system may promote the availability and convenience of DIF, which has a reverse causal effect. Therefore, DIF as the core explanatory variable may have a certain degree of endogeneity. In order to solve regression error problem caused by endogeneity, this paper adopts the instrumental variable method to more accurately identify the causal relationship between them. The instrumental variables should select the variable with historical or natural attributes, and the selected variable must be independent of the explained variable but related to the explanatory variable. Therefore, this paper selects the number of highway mileage in each city as an instrumental variable. Firstly, the number of highway mileage has certain geographical attributes. Secondly, there is a certain correlation between highway mileage and FIF. In fact,highway is one of the ways of information transmission, highway condition not only determines the communication and development between the region and the outside world, but also determines the degree of transmission of digital finance by the government and enterprises. Highway mileage, as a instrumental variable, conforms to the hypothesis related to endogenous variables. The instrumental variable regression model is as follows:

$$DIF_{it} = \alpha_1 + \beta_1 Railway_{it} + \delta_1 X_{it} + \mu_i + \varepsilon_{it} \tag{5}$$

$$OIS_{it} = \varphi_0 + \varphi_1 DIF_{it} + \eta_1 X_{it} + \mu_i + \varepsilon_{it} \tag{6}$$

$$IS_{it} = \gamma_0 + \gamma_1 DIF_{it} + \eta_2 X_{it} + \mu_i + \varepsilon_{it} \tag{7}$$

Eq (5) represents the extent to which DIF is explained by railway, Eq (6) denotes the degree of influence of DIF on OIS, Eq (7) represents the degree of influence of DIF on RIS. As shown in the following table, instrumental variables have a certain degree of explanation for endogenous variables. F test shows that IV is significant at 1% significance level, and there is no problem of weak instrumental variables. After introducing instrumental variables, as shown in Table 6, the column (19) is listed as the regression result of OIS, the column (20) is listed as the regression result of RIS. DIF still has the effect of promoting IT, if DIF increases by 1%, OIS will increase by 0.28% and RIS will decrease by 0.610%. The empirical results show that DIF still has a certain role in promoting industrial upgrading after eliminating the endogenous influence.

## 5.4 Mediating effect

Digital inclusive finance is a leading industry. The flow and allocation of financial specialty will drive the flow of social capital, labor, land and other production factors. Therefore, the efficient allocation of DIF will promote the efficient allocation of other social resources. There is no doubt that the efficient allocation of resources, diversity, convenience and low cost of resource allocation will promote the growth of residents' income and the improvement of innovative technologies. On the other hand, the growth of residents' income will promote the upgrading of residents 'consumption and consumption growth. The diversification of residents' consumption and the upgrading of consumption level will promote the rapid development of industries, especially the tertiary industry, and then promote the upgrading of urban industries [55]. At the same time, the promotion of DIF has improved the financial

**Table 6. Estimation results of instrumental variable method.**

|  | (19) | (20) |
|---|---|---|
| DIF | 0.280***(16.01) | -0.610***(-7.81) |
| DR | -0.008(-0.80) | 0.050(1.11) |
| RO | 0.006(1.30) | -0.036*(-1.84) |
| ED | -0.425(-0.81) | -3.731(-1.59) |
| CI | 1.113(0.21) | 34.679(1.44) |
| PD | -0.165**(-2.09) | 0.739**(2.09) |
| N | 1035 | 1035 |
| $R^2$ | 0.572 | 0.132 |
| First stage IV | 0.076**(2.33) | 0.076**(2.33) |
| Contral variable | Yes | Yes |
| F test | 11.94*** | 72.83*** |

Notes

* $p < 0.1$

** $p < 0.05$

*** $p < 0.01$ the bracket represents the t value.

environment for enterprise innovation and entrepreneurship. Through the mitigation mechanism of credit constraints and the improvement mechanism of social trust, the financial availability and financing efficiency of small and medium-sized enterprises are improved, and the innovation financing threshold and financing cost are reduced, so as to release the innovation ability of scientific and technological active subjects, such as small and medium-sized enterprises, and accelerate IT. To test this mechanism, this paper chooses the average on-job salary to represent the income level of residents, the number of invention patents to represent the level of technological innovation, and the income level of residents and technological innovation as the mediating variables. Using the method of Baron and Kenny [65], this paper uses recursive model to test the transmission mechanism of DIF affecting IT through residents' income level and technological innovation. The mediating effect model is expressed as:

$$Patent_{it} = \alpha_0 + \theta_0 DIF_{it} + \lambda_0 Z_{it} + \mu_i + \varepsilon_{it} \tag{8}$$

$$OIS_{it1} = \vartheta_0 + \varphi_0 DIF_{it} + f_0 Patent_{it} + \lambda_0 Z_{it} + \mu_i + \varepsilon_{it} \tag{9}$$

$$RIS_{it2} = \vartheta_1 + \varphi_1 DIF_{it} + f_1 Patent_{it} + \lambda_1 Z_{it} + \mu_i + \varepsilon_{it} \tag{10}$$

$$Wage_{it} = \alpha_1 + \theta_1 DIF_{it} + \lambda_1 Z_{it} + \mu_i + \varepsilon_{it} \tag{11}$$

$$OIS_{it2} = \vartheta_2 + \varphi_2 DIF_{it} + f_2 Wage_{it} + \lambda_2 Z_{it} + \mu_i + \varepsilon_{it} \tag{12}$$

$$RIS_{it2} = \vartheta_3 + \varphi_3 DIF_{it} + f_3 Wage_{it} + \lambda_3 Z_{it} + \mu_i + \varepsilon_{it} \tag{13}$$

Firstly, this paper examines the impact of DIF on structural transformation. Secondly, it estimates Eq (8) to test whether the impact of DIF on technological innovation is significantly positive or not. If it is significantly positive, it means that DIF will promote the new improvement of technological innovation. Finally, it performs regressions on Eqs (9) and (10) respectively. If the two coefficients and f are significant, and the coefficient φ of DIF is lower than the coefficients of Eqs (1) and (4), it indicates that technological innovation has a partial mediating

**Table 7. Estimated results of mediating variable method.**

| | OIS | | | | RIS | | | |
|---|---|---|---|---|---|---|---|---|
| | Patent | OIS | Wage | OIS | Patent | RIS | Wage | RIS |
| Patent | | 0.055*** (6.57) | | | | -0.072* (-1.88) | | |
| Wage | | | | 0.380*** (11.44) | | | | -0.290* (-1.80) |
| DIF | 0.822*** (24.83) | 0.179*** (16.45) | 0.424*** (52.91) | 0.063*** (3.89) | 0.856*** (26.49) | -0.395*** (-7.86) | 0.432*** (56.35) | -0.333*** (-4.23) |
| Contral | yes | yes | yes | yes | yes | yes | yes | Yes |
| $R^2$ | 0.451 | 0.609 | 0.794 | 0.642 | 0.448 | 0.140 | 0.790 | 0.143 |
| N | 1035 | 1035 | 1035 | 1035 | 1035 | 1035 | 1035 | 1035 |

Notes

* $p<0.1$

** $p<0.05$

*** $p<0.01$ the bracket represents the t value.

effect in the impact of DIF on industrial upgrading. If the parameter estimations of $\varphi_0 \varphi_1$ do not pass the significance test and f is significant, it indicates that technological innovation has a complete mediating effect in the impact of DIF on industrial upgrading. Similarly, the effect mechanism of residents' income is the same.

The regression results of DIF to IT have been reported in Table 3, this part is not listed in Table 7. Table 3 shows that no matter the optimization or rationalization of industrial structure, DIF has a significant impact on IT. The regression results of the above formula are shown in Table 7. In the regression of industrial structure upgrading, DIF has a significant positive impact on technological innovation and residents' income. After introducing intermediate variables, as shown in Table 7, the DIF coefficient is reduced, and the intermediary effect of DIF is significant at 1% significance level, indicating that technological innovation and residents' income have a significant partial intermediary role in the impact of DIF on OIS, which verifies the hypothesis (2). When regressing RIS, DIF has a positive and significant impact on the intermediary variables. After adding intermediary variable to perform regression analysis, both DIF and intermediary variables are significant, and the impact of DIF on RIS is significantly reduced, indicating that technological innovation and residents' income also have a significant partial intermediary role in the impact of DIF on industrial structure rationalization, which verifies the hypothesis (3).

### 5.5 Difference analysis

In view of the mediating effect of DIF on IT, this paper categorizes industrial transformation based on intermediary indicators and conducts difference analysis. Although DIF adheres to the service concept of benefiting everyone, it may have significant differences in income levels of residents. Due to long-term existence of dual economy in resource-based cities, in aspect of factor endowments of residents' income, income distribution, education level and availability of financial services exist huge difference. Especially, in high-income areas, the level of financial services and economic development are higher, and residents have stronger acceptance of DIF, it is easier for them to widely enjoy financial expertise and services, thus directly upgrading urban consumption habits and industrial structure.

According to the median of residents' income in 2011, the sample is divided into high income level and low-income level to conduct regression analysis. As shown in Table 8, the

**Table 8. Income level heterogeneity.**

|  | (21) | (22) | (23) | (24) |
|---|---|---|---|---|
| DIF | 0.235***(19.50) | 0.210**(17.03) | -0.419***(-7.74) | -0.508***(-9.11) |
| DR | -0.004(-0.36) | -0.010(-0.59) | 0.058(1.05) | -0.005(-0.06) |
| RO | 0.001(0.10) | 0.001(1.50) | -0.056**(-2.00) | -0.012(-0.43) |
| ED | -1.598(-1.52) | -0.730(-1.14) | 1.860(0.41) | -1.098(-0.38) |
| CI | 12.511(1.19) | 3.456(0.51) | -16.643(-0.37) | 6.932(0.23) |
| PD | -0.517**(-2.50) | -0.055(-0.66) | -0.474**(-2.17) | 0.059(0.30) |
| $R^2$ | 0.588 | 0.602 | 0.137 | 0.160 |
| Contral | yes | yes | yes | yes |
| N | 513 | 522 | 513 | 522 |

Notes

* $p < 0.1$

** $p < 0.05$

*** $p < 0.01$ the bracket represents the t value.

column (21) and (22) are listed as the regression results of OIS to low income and high-income groups. The results show that in the regression of OIS, no matter in the high-income group or the low-income group, DIF has a positive effect on IT. However, the regression coefficient for the low-income group is 0.235, which is significant at the 1% significance level. The regression coefficient of the high-income group is 0.210, and the influence coefficient of the high-income group is lower than the influence coefficient of the low-income group. This is because the resources in low-income areas tend to be depleted, and the original resource-based industries are not enough to maintain development. Therefore, the promotion of DIF in low-income cities is more likely to stimulate the development of the service industry. The results in columns (23) and (24) show the regression results of industrial rationalization on the low-income and high-income groups. Since most high-income cities are coastal cities with convenient transportation and natural trade conditions, they form abundant capital deposit. With the development advantages, industrial transformation in high-income cities is reasonable, and it is easier to restrain the development of extractive industries. Therefore, DIF has a greater impact on IT in high-income group than in low income group.

As an inclusive financial resource, DIF can provide credit support for potential entrepreneurs who lack mortgage assets, and can also enrich entrepreneurial information and resources through digital platforms. Due to the rise of digital technologies, such as big data and cloud computing, and the development of digital platforms is open and boundless, it reduces the learning cost and prior cost of technological innovation. It is obvious that technological innovation is the main driving force of IT, therefore, this paper divides the samples into high innovation level and low innovation level through the median of technological innovation. In Table 9, the column (25) and (26) are listed as the regression results of OIS to low innovation and high innovation groups, the column (27) and (28) are listed as the regression results of RIS to low innovation and high innovation groups. Innovation is the recombination of production factors, during this process, production factors flow across sectors due to the pursuit of economic benefits. From sectors with slow productivity to the sectors with rapid productivity growth, this process corrects the previous distorted industrial structure to promote the transformation and upgrading of industrial structure, so as to realize industrial extension and solve the problem of resource dependence. Therefore, no matter the regression of OIS or the regression of RIS, DIF in high innovation areas has a greater impact on IT and upgrading, which verifies the impact of innovation differentiation.

**Table 9. Heterogeneity of innovation level.**

| | (25) | (26) | (27) | (28) |
|---|---|---|---|---|
| DIF | 0.218***(18.71) | 0.227***(18.20) | -0.428***(-4.37) | -0.485***(-9.73) |
| DR | 0.001(0.03) | -0.029(-1.59) | -0.008(-0.14) | 0.219***(2.94) |
| RO | -0.003(-0.47) | 0.017**(2.49) | -0.032(-1.18) | -0.035(-1.29) |
| ED | -0.202(-0.29) | -2.018**(-2.52) | 3.087(0.95) | -6.152*(-1.93) |
| CI | -1.868(-0.27) | 17.776**(2.11) | -30.586(-0.92) | 61.802*(1.13) |
| PD | -0.751***(-4.29) | 0.067(0.76) | -0.146(-0.63) | 0.264(-1.73) |
| $R^2$ | 0.628 | 0.581 | 0.126 | 0.194 |
| Contral | yes | yes | yes | yes |
| N | 513 | 522 | 513 | 522 |

Notes

* $p<0.1$

** $p<0.05$

*** $p<0.01$ the bracket represents the t value.

# 6 Discussion

Through the panel regression model, it verifies that DIF has a positive effect on the optimization of industrial structure and a negative effect on the rationalization of industrial structure, indicating that DIF has accelerated the transition to productive service industry and reduced the proportion of manufacturing. This is mainly because digital industrialization and industrial digitalization are promoting the high integration of digital finance and traditional finance, making the social and industrial forms become more diversified. In the process of continuous improvement of the digital financial system to enhance production effectiveness and change business models, the third industry-oriented business model has been recognized as a sign of industrial success [66]. The changes in social demand and supply affect the production of the industrial chain. According to the principle of "consumer rule", consumers maximize their economic interests through "currency votes", and pursue emerging technologies and high-tech products. Driven by the agglomeration effect, the geographical and spatial layout of various industries changes. New-type enterprises gradually invade the market, thus realizing a decline in the proportion of manufacturing and an increase in the proportion of service industries. For example, Ordos city is a mature city among resource-based cities. Relying on internet financial promotion, it aims to actively promote the development of non-coal industry, steadily push the development of automobile manufacturing, electronic information and other industries according to market preferences, finally, it successfully promotes the high-end resource industries and the parallel development of new industries. In the meanwhile,with the continuous improvement of the industrial chain, the government also issues a series of policies to ensure the smooth progress of IT. The State Council successively promulgated *Guidance on Accelerating the Development of Producer Services to Promote the Adjustment and Upgrading of Industry and Structure* and *Opinions on Supporting Industrial Transformation and Upgrading of Old Industrial Cities and Resource-based Cities*. Therefore, the government should pay more emphasis on the coordination of digital inclusive finance and industrial structure, optimize the internal structure of the industry through digital means, attract more high-tech labor, form a virtuous circle within the industry, and make it play a "follow-up power" to improve the efficiency of industrial transformation.

Residents 'income is the mediating effect of digital inclusive finance on industrial transformation. The continuous growth of residents' income is an important factor for the stable and

high-quality development of a country. Digital inclusive finance mainly affects residents 'income by influencing three major effects, namely, the threshold effect, the exclusion effect and the poverty reduction effect. Obviously, it solves the problem of credit constraint of financial exclusion group, and changes the cost and efficiency of reaching users. The absolute income theory, proposed by Keynes, believes that consumption will increase as income increases. Therefore, the increase in residents' income promotes the consumption orientation, and the diversification of consumption demand and the upgrading of consumption level will promote the transformation of industrial structure, thus driving the transformation of urban industry. The government implemented *G20 Advanced Principles of Digital inclusive Finance* and *White Paper on Digital Inclusive Finance* to achieve inclusive and sustainable growth of residents ' income and industrial economy. Digital inclusive financial technologies, such as home-based fast-loan apps, general-benefit apps and Ant Financial Services have been well practiced in Henan Province. At present, there are hundreds of thousands of people registered general-benefit apps in Henan Province. The cumulative number of processed loans is over 17,000, and the amount of loans applied exceeds CNY 2.15 billion. It gives full play to the role of optimal allocation of resources, and improves the income level of residents, thus promoting the industrial development in Henan Province.

Technological innovation is the mediating effect of digital inclusive finance on industrial transformation. Digital inclusive finance, relying on its inherent advantages of digital technology and inclusive finance, provides financial products and services to "corporate long-tail groups" in digital form, which solves the financing problem of enterprises and provides solutions for promoting R&D and innovation. Digital inclusive finance lowers the financing threshold for enterprises and broadens their financing channels. Under the big data of information disclosure, the market implements precise regulatory policies, thereby reducing the problem of information asymmetry and improving financing efficiency. With the help of online trading platform, digital inclusive finance has increased the speed of loan approval by 20%, it is obvious that such fast and efficient method of loan review reduces the financing cost of technological innovation and stimulates the driving force of technological innovation. Meanwhile, digital inclusive finance stimulates technological innovation by improving fiscal instruments and tax policy accuracy. In addition, technology is one of the three elements of industrial production. Therefore, digital inclusive finance improves the level of industrial transformation by improving the level of technological innovation. In recent years, Guangzhou Province introduced *Guangzhou's implementation opinions on promoting the development of supply chain finance* and *Guangzhou City to implement supply chain financial services for small and medium-sized enterprises innovation pilot implementation plan*, it is obvious that Guangzhou focuses on emerging industries and characteristic areas to carry out innovation pilot, which lead to the rapid development of industrial economy in Guangzhou.

The World Bank divides inequality into inequality of outcome and inequality of opportunity. The inequality of outcome mainly includes inequality of wealth and income, and inequality of opportunity mainly includes education and other non-income gaps. Therefore, in the case of inequality of residents ' income and technological innovation technology, the effect of digital inclusive finance on industrial transformation is different. For example, high-income cities are relatively rich in resources, with certain resource endowments, capital deposits and development potential. Moreover, high-income cities are mostly located in the eastern region. The eastern region has advantages in geographical development and diversified industrial forms, which makes it easier to restrain the development of extractive industries. Therefore, in terms of the optimization of industrial structure, the effect of high income is more obvious. However, due to the exhaustion of resources in low-income areas and the pressure to upgrade the industrial structure, coupled with the law of diminishing marginal benefits of capital, low-

income areas are driven by digital inclusive finance and the tertiary industry has developed rapidly. For high-tech regions, the development advantages are distinct and there are many channels for industrial transformation, inevitably, the effect of industrial transformation is significant. In the innovation group, the industrial extension brought by high innovation makes urban industries gradually get rid of resource dependence and realize industrial diversification. High-innovation cities are more likely to accept digital inclusive finance to promote industrial transformation.

## 7 Conclusions and policy recommendations

Firstly, this paper analyzes the impact of digital inclusive finance on industrial transformation, taking 115 resource-based cities in China from 2011 to 2019 as the research objects. The empirical results show that digital inclusive finance and its decomposition term have a positive pulling effect on industrial transformation. Secondly, this paper introduces the variable of residents' income and technological innovation, and it tests the establishment of intermediary transmission. Finally, taking the median of residents' income and technological innovation as the dividing line, it conducts heterogeneity analysis on high and low income and high and low innovation groups. The results show that for the rationalization of industrial structure regression, the effect of industrial transformation in low-income areas is better, and for the optimization of industrial structure, the efficiency of manufacturing upgrading in high-income areas is better, resulting in more significant industrial transformation. In high innovation cities, whether the optimization of industrial structure or the rationalization of industrial structure, digital inclusive finance has a better effect on industrial transformation.

Based on theoretical analysis and empirical research, this paper puts forward the following policy recommendations: (1) Digital inclusive finance is an important influence factor in industrial transformation, the government should increase policy support for digital inclusive finance, increase government subsidies and tax rebate systems, leading to "rent-seeking behavior" of enterprises. (2) The government should intensify financial reform, increase information transparency, increase the availability and inclusiveness of digital inclusive finance, accelerate the integration of digital finance and industrial economy, and create a "wide coverage, multi-level and efficient" digital financial inclusive system to promote the improvement of the internal system of the industry. (3) According to the happiness index and development level of different regions, it is necessary to implement the targeted industrial transformation scheme. For example, high-income regions should improve the effectiveness and service quality of digital financial services, while low-income regions should continue to promote the infrastructure construction of digital financial services, expand the coverage of digital finance, increase support policies for low-income vulnerable groups, and solve the problem of "full coverage"of financial services. In addition, high innovation city should actively promote the initiative of innovation, continue to promote the research and development of financial products as well as innovation of service mode, while low-innovation cities need to maintain communication with high innovation cities to avoid the problem of digital divide and thus generate technological exclusion.

## Supporting information

**S1 Data.**
(XLSX)

## Author Contributions

**Conceptualization:** Fei Li.

**Data curation:** Jinli Liu.

**Formal analysis:** Jinli Liu.

**Funding acquisition:** Jinli Liu.

**Investigation:** Yufei Wu.

**Methodology:** Yufei Wu.

**Project administration:** Yufei Wu.

**Resources:** Yufei Wu.

**Software:** Shen Zhong.

**Supervision:** Shen Zhong.

**Validation:** Shen Zhong.

**Visualization:** Shen Zhong.

**Writing – original draft:** Fei Li.

**Writing – review & editing:** Fei Li.

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
