## [Decision Letter · Decision Letter 0]

27 Jun 2022

PONE-D-22-09445Does digital inclusive finance promote industrial transformation? New evidence from 115 resource-based cities in ChinaPLOS ONE

Dear Dr. Zhong,

Thank you for submitting your manuscript to PLOS ONE. After careful consideration, we feel that it has merit but does not fully meet PLOS ONE’s publication criteria as it currently stands. Therefore, we invite you to submit a revised version of the manuscript that addresses the points raised during the review process.

We look forward to receiving your revised manuscript.

Kind regards,

Alessandro Margherita

Academic Editor

PLOS ONE

Journal Requirements:

Reviewers' comments:

Reviewer's Responses to Questions

**Comments to the Author**

1. Is the manuscript technically sound, and do the data support the conclusions?

Reviewer #1: Yes

Reviewer #2: Yes

2. Has the statistical analysis been performed appropriately and rigorously? 

Reviewer #1: Yes

Reviewer #2: I Don't Know

3. Have the authors made all data underlying the findings in their manuscript fully available?

Reviewer #1: No

Reviewer #2: No

4. Is the manuscript presented in an intelligible fashion and written in standard English?

Reviewer #1: Yes

Reviewer #2: Yes

5. Review Comments to the Author

Reviewer #1: The article is remarkable, the presentation is good and the article has provided informative material for the investors and academic researchers. The research is very helpful for Chinese researchers as well.

Reviewer #2: Enhance discussion of theoretical/academic advancements against extant literature.

Include more works of 2021 and 2022 and reinforce the research gap and the positioning of the paper on the new literature.

6. PLOS authors have the option to publish the peer review history of their article (what does this mean?). If published, this will include your full peer review and any attached files.

Reviewer #1: No

Reviewer #2: No

---

## [Author Response · Author response to Decision Letter 0]

17 Jul 2022

Figure 2 in our submission contain [map/satellite] images which may be copyrighted. Therefore, we remove Figure 2 from our submission.

---

## [Editor Report · Decision Letter 1]

15 Aug 2022

Does digital inclusive finance promote industrial transformation? New evidence from 115 resource-based cities in China

PONE-D-22-09445R1

Dear Dr. Zhong,

We’re pleased to inform you that your manuscript has been judged scientifically suitable for publication and will be formally accepted for publication once it meets all outstanding technical requirements.

Kind regards,

Alessandro Margherita

Academic Editor

PLOS ONE
---

## [Editor Report · Acceptance letter]

18 Aug 2022

PONE-D-22-09445R1 

Does digital inclusive finance promote industrial transformation? New evidence from 115 resource-based cities in China 

Dear Dr. Zhong:

I'm pleased to inform you that your manuscript has been deemed suitable for publication in PLOS ONE. Congratulations! Your manuscript is now with our production department. 

Kind regards, 

on behalf of

Dr. Alessandro Margherita 

Academic Editor

PLOS ONE